# Machine Learning Techniques for the Prediction of the Magnetic and Electric Field of Electrostatic Discharges

**Georgios Fotis, Vasiliki Vita and Lambros Ekonomou \***

Department of Electrical and Electronics Engineering Educators, ASPETE—School of Pedagogical and Technological Education of Athens, 14121 Heraklion, Greece; gfotis@gmail.com (G.F.); vasvita@aspete.gr (V.V.)
\* Correspondence: leekonomou@aspete.gr; Tel.: +30-210-2896-955

**Abstract:** The magnetic and electric fields of electrostatic discharges are assessed using the Naïve Bayes algorithm, a machine learning technique. Laboratory data from electrostatic discharge generators were used for the implementation of this algorithm. The applied machine learning algorithm can be used to predict the radiated field knowing the discharge current. The results of the Naïve Bayes algorithm are compared to a previous software tool derived by Artificial Neural Networks, proving its better outcome. The Naïve Bayes algorithm has excellent performance on most classification tasks, despite its simplicity, and usually is more accurate than many sophisticated methods. The proposed algorithm can be used by laboratories that conduct electrostatic discharge tests on electronic equipment. It will be a useful software tool, since they will be able to predict the radiating electromagnetic field by simply measuring the discharge current from the electrostatic discharge generators.

**Keywords:** electrostatic discharge generators; electrostatic discharge current; contact discharges; electromagnetic field; machine learning; Naïve Bayes algorithm

## 1. Introduction

The sudden transfer of charge between objects at different potentials is defined as Electrostatic Discharge (ESD). The "triboelectric effect" is a phenomenon where materials can develop electrostatic charge when they are separated from a different material with which they were in contact with [1]. The peak discharge current may have a value of a few amperes during discharge. Consequently, ESD can cause malfunction or destruction to electronic devices or microelectronic circuits, although the phenomenon is extremely short in duration [2].

Electrostatic discharges have become a significant source of not only electromagnetic (EM) interference but also of physical damage for modern electronics, with charging voltages reaching tens of kV and discharge periods of less than a few ns [3,4]. ESD becomes progressively more critical when technology advances and greater working frequency technologies are employed due to its high frequency properties. At frequencies above 1 GHz, sub-ns rise time discharges produce radiated fields and generated disturbances with a considerable spectrum component, making circuits operating at these frequencies particularly vulnerable to damage or performance deterioration. Manufacturers and product designers have always been concerned about these phenomena, which has led to a substantial amount of research and standardization effort on the precise and repeatable simulation of ESD [5–10].

Standard IEC 61000-4-2 [11] defines the testing procedure on equipment that contains electrical or electronic circuits against ESD. This equipment is known in the EMC labs as Equipment Under Test (EUT), and its ESD test results may vary for different ESD generators, although during the tests, the charging voltage and the contact discharge current of these generators may be the same. Moreover, the different orientation of the same ESD generator may lead to different test results. It must be mentioned that during the verification of the

ESD generators, ESD discharges are contact discharges because the reproducibility of air discharge currents is a major problem [12–15].

The first attempt for simultaneously measuring the produced EM field and current during ESD was by Ma and Wilson [16]. After them, Pommerenke [17] measured the produced EM field for both contact and air discharges at a distance between 0.1 and 1 m. In [18,19], the waveform of the ESD current can be calculated from measurements of the EM field. Researchers have also designed an ESD detection system by using relevant EM fields [20]. The EM field during ESD was measured at the calibration setup in [21] and [22], demonstrating that field measurement is a difficult operation with results that vary based on the construction details of the EM field probes and the measurement setup. Recent research on the produced EM field radiating by ESD for various laboratory setups has been conducted [23–26], providing a better base in the selection of ESD test levels in comparison to actual ESD levels.

For EM field measurements [27,28] relative to Pellegrini targets, a current transducer is placed at a metal plane's center and it showed that there is a different EM field not only for different ESD generator models but also for the same ESD generator depending on its orientation. Working in this direction and trying to describe the measured current with high accuracy, there have been studies where different optimization methods have been developed for the optimum parameter calculation of the ESD current's equation [29,30].

In this paper, a machine learning technique is applied. Machine learning (ML) [31,32] is an area of artificial intelligence (AI) that focuses on using data and algorithms to mimic the way humans learn, with the goal of steadily improving accuracy. ML is significant because it allows businesses to see trends in customer behavior and business operating patterns while also assisting in the development of new products. In this research study, the Naïve Bayes algorithm (NBA) is used for assessing the EM field radiated by ESD. The application of NBA for classification has received increased attention. The algorithm itself has its roots in pattern recognition [33]. The work presented in [34] addresses its drawbacks and compares it to a learning algorithm of instance base structure. Its remarkably high precision has been highlighted to other sophisticated learning approaches [31,32,35]. In [36], NBA compared to state-of-the-art algorithms for decision tree induction, instance-based learning, and rule induction and it was observed that NBA was be superior to them.

In [15], NBA was used to predict the rise time and the maximum current using as input data humidity and the voltage before discharge. In the current study, NBA is used as a prediction tool for EM fields produced by ESD generators from the discharge current's characteristics in terms of the distance and the direction of the ESD generator from the tested equipment. These data sets are measurements of both the ESD current and the EM field generated by ESD. The NBA results are compared to previous work [27,28] in which ANN had been used instead. This work proposes a machine learning method that can be a very useful tool for laboratories conducting ESD tests [11].

## 2. The IEC 61000-4-2

According to [11], every ESD generator must produce a discharge current, as indicated in Figure 1, according to the Human Body Model (HBM). The pulse in Figure 1 contains two peaks: an "initial peak" induced by the hand discharge and a second lower peak caused from the body's discharge. The rise time ($t_r$) of the first peak is 0.8 ns ($\pm$25%), and its amplitude is determined by the ESD simulator's charging voltage.

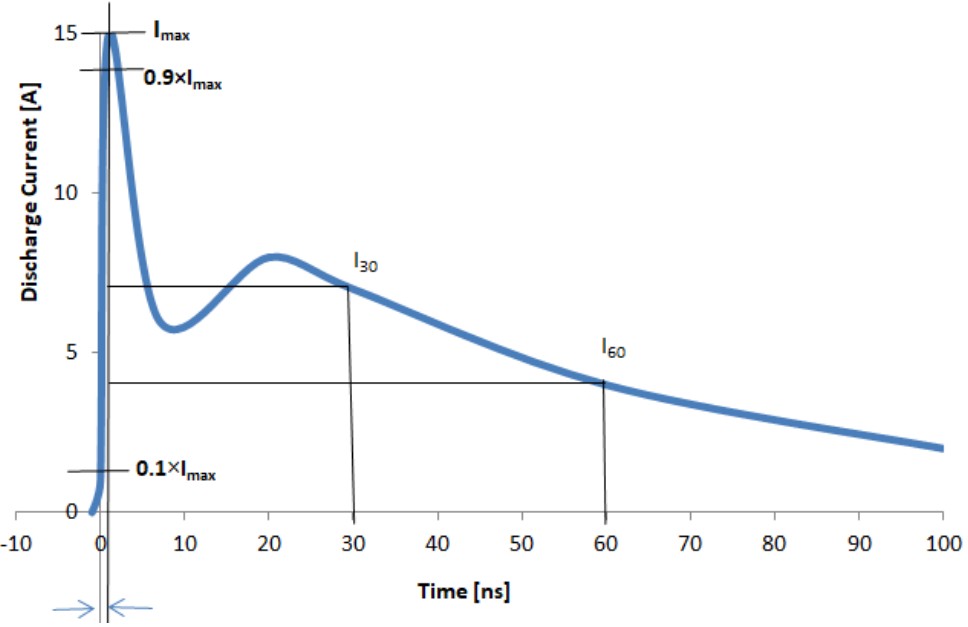

**Figure 1.** Typical waveform of the ESD current [11].

There are four parameters for which its values must be constrained by predefined limitations according to [11]: $t_r$, $I_{max}$, $I_{30}$, and $I_{60}$. Currents $I_{30}$ and $I_{60}$ are determined for 30 ns and 60 ns, respectively, as shown in Figure 1. These parameter limits, presented in Table 1, concern only contact discharges.

**Table 1.** Parameters of the ESD current [11].

| Level | Charging Voltage (kV) | $I_{max}$ (A) | Accepted Deviation | $t_r$ (ns) | Accepted Deviation | $I_{30}$ (A) | Accepted Deviation | $I_{60}$ (A) | Accepted Deviation |
|---|---|---|---|---|---|---|---|---|---|
| 1 | 2 | 7.5 | | | | 4 | | 2 | |
| 2 | 4 | 15 | ±15% | 0.8 | ±25% | 8 | ±30% | 4 | ±30% |
| 3 | 6 | 22.5 | | | | 12 | | 6 | |
| 4 | 8 | 30 | | | | 16 | | 8 | |

## 3. Current and EM Field Measurements

### 3.1. Laboratory Setup

In Figure 2, the laboratory setups for the EM field measurement are presented. The discharge current and the radiating field (electric or magnetic) produced by contact discharges were monitored simultaneously using a TDS 7254B oscilloscope. NSG-433 and NSG-438 ESD generators of Schaffner were used. Throughout the experiment, the high voltage cable was held in the same place. A resistive load (Pellegrini target) [11] was placed in the center of a metal plane for measuring the ESD current.

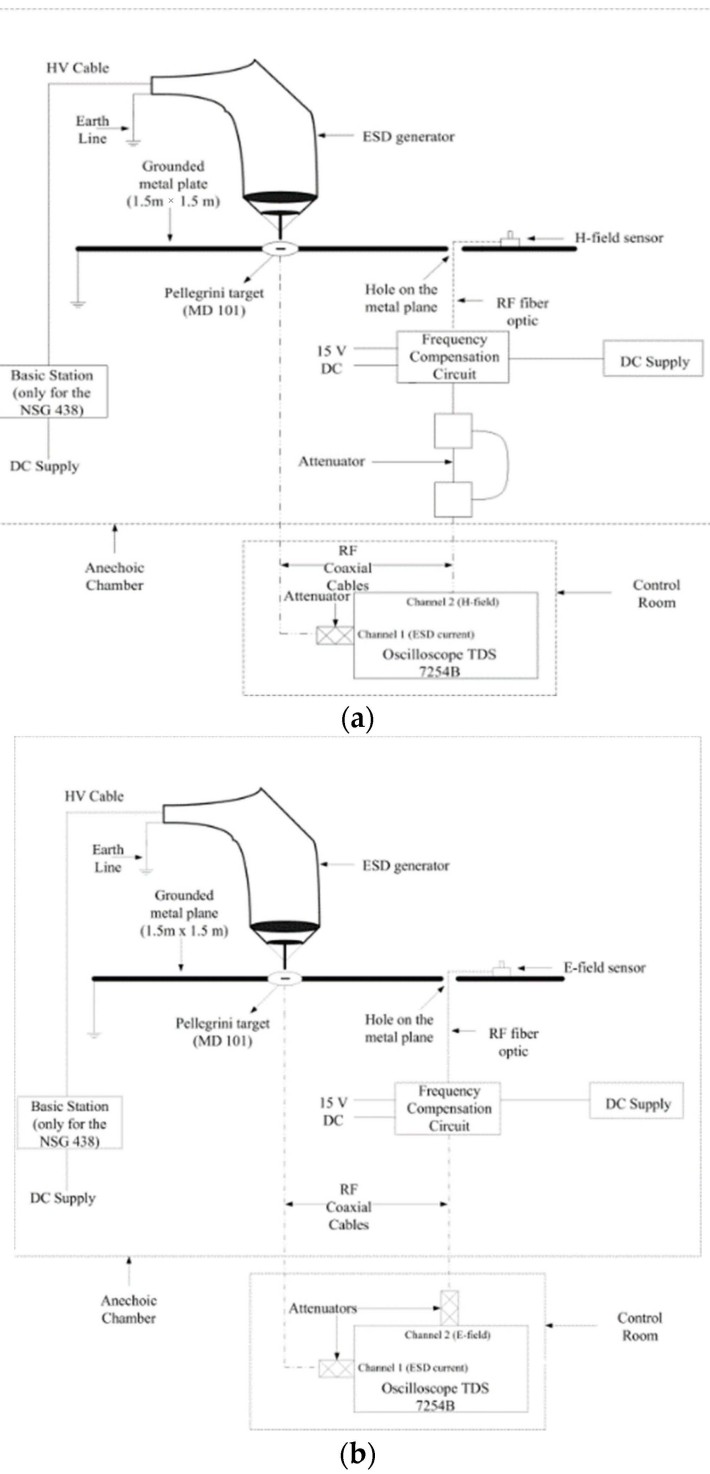

**Figure 2.** Laboratory setups: (**a**) measurement of the H-field and (**b**) measurement of the E-field.

The sensors used were designed by Professor Pommerenke [37] and they were at different positions from the Pellegrini target and in three different directions, as shown in Figure 3. Due to interference from the ESD generator's ground strap, measurements in direction B were not performed. The ground strap was at a distance of 1 m from the discharge point, as defined by [11], and the loop had a big enough curvature to reduce the uncertainty in the measurement of the H-field.

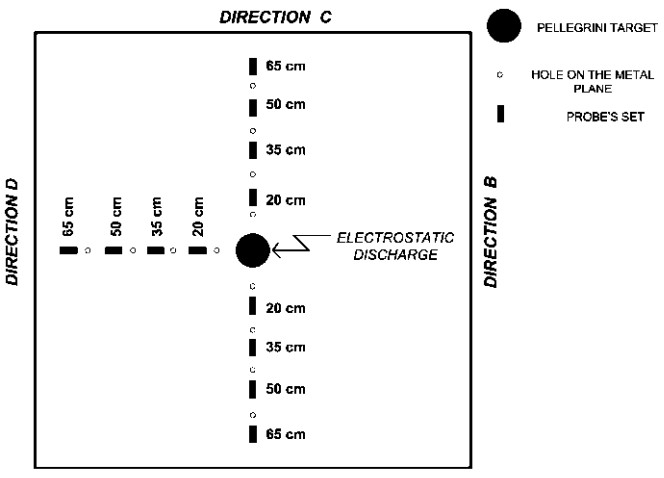

**Figure 3.** The measurement points where the field sensors were placed.

*3.2. Laboratory Results*

Figure 4a,b show typical measurement samples of the EM field with a simultaneous measurement of the ESD current for the NSG-438 ESD generator. The field sensor was 20 cm from the Pellegrini target in direction A. Figure 5a,b show the maximum H-field and E-field for the two used ESD generators for a charging voltage of +2 kV in direction D. Each generator radiates its unique EM field. In Figure 6a,b, the absolute maximum value of the EM field for the NSG-438 ESD generator in all three orientations is depicted, demonstrating that the radiating EM field varies based on the ESD generator's orientation.

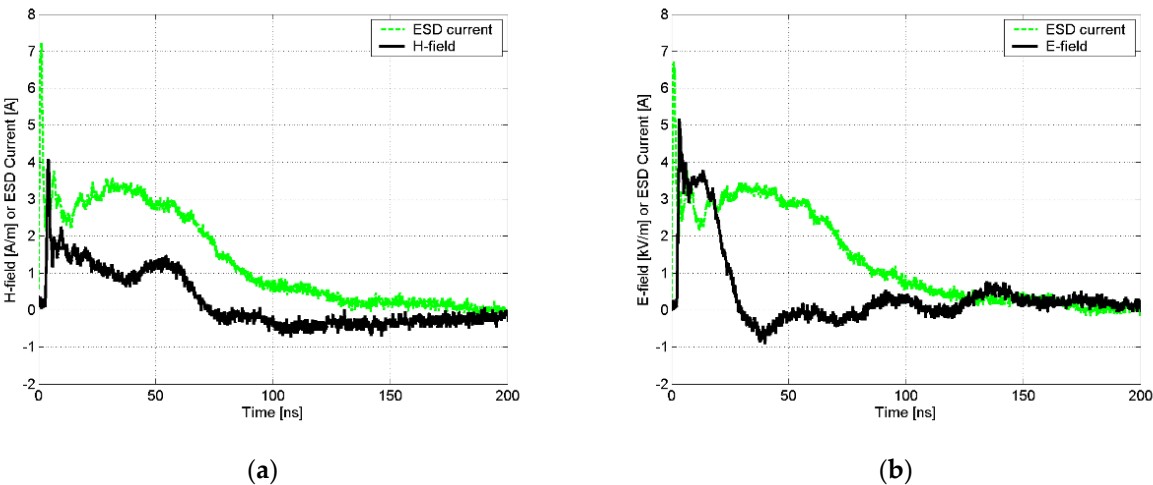

(**a**)　　　　　　　　　　　　　　　　　　　　　　　　　(**b**)

**Figure 4.** Measurements in direction A, 20 cm from the Pellegrini target for the NSG−438 (charging voltage = +2 kV): (**a**) discharge current and magnetic field and (**b**) discharge current and electric field.

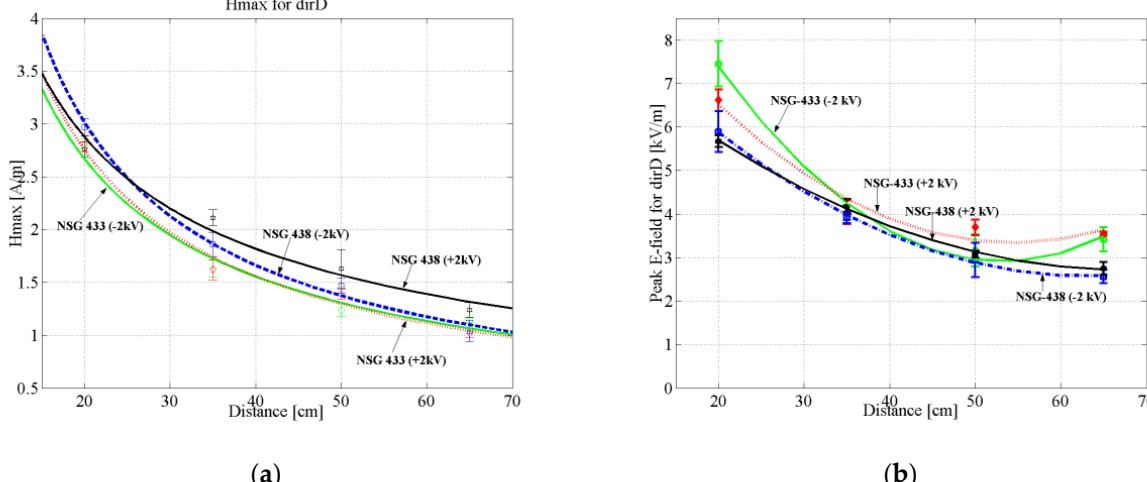

**Figure 5.** Measurements of the two ESD generators for various distances in direction D (charging voltage = +2 kV): (**a**) maximum magnetic field and (**b**) maximum electric field.

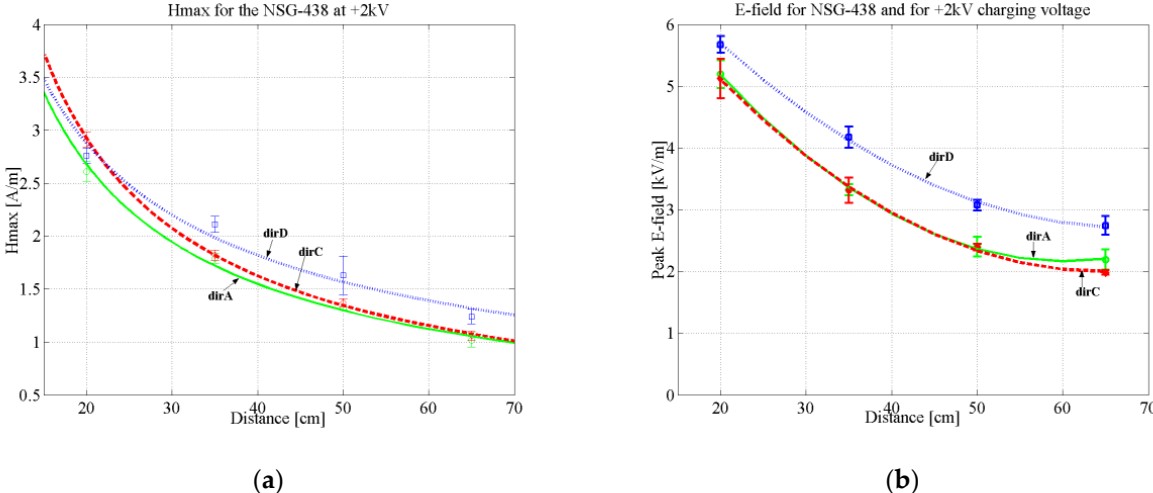

**Figure 6.** Measurements of the NSG-438 ESD generator in all three directions (charging voltage = +2 kV): (**a**) maximum magnetic field and (**b**) maximum electric field.

## 4. Machine Learning

Machine learning (ML) is a research field concerned with comprehending and developing methods that 'learn', using data to improve performance on a set of tasks [38]. It is considered an artificial intelligence component. ML algorithms use training data to form a model for predictions without the need for explicit programming. ML algorithms are used almost in any science and human activity, such as medicine, image recognition, and email filtering and where other algorithms and techniques are difficult to implement [39]. Despite the advances in ML in the last years, it has been proven [31,32,35,36] that NBA is not only simple but also fast, accurate, and reliable. Moreover, this algorithm has been implemented in various science fields, proving its efficiency [40–44].

### 4.1. Learning Classifiers—Bayes Rule

In a learning problem, the approximation of target function f: $P(Y \mid X)$ is needed. Y is a Boolean random variable and $X = (X_1, X_2, \ldots, X_n)$, where $X_i$ is the Boolean random variable [38].

Bayes rule $P(Y = y_i \mid X)$ can be written as follows.

$$P(Y = y_i | X = x_k) = \frac{P(X = x_k | Y = y_i)\ P(Y = y_i)}{\sum_j P(X = x_k | Y = y_i)\ P(Y = y_i)} \tag{1}$$

$x_k$ is the $k$th possible vector value for X, $y_m$ is the $m$th possible value for Y, and the denominator's sum is the overall legal values of the random variable Y.

Using the training data for the estimation of $P(X \mid Y)$ and $P(Y)$, $P(Y \mid X)$ is learned. Then, we can use these estimates for any new instance $x_k$ with the Bayes rule for the determination of $P(Y \mid X = x_k)$.

### 4.2. Naïve Bayes Algorithm

The Naïve Bayes classifier lowers the Bayesian classifiers complexity by assuming conditional independence, which decreases the estimated number of parameters.

### 4.2.1. Conditional Independence

**Definition 1.** *If X, Y, and Z are random variables sets, and X is conditionally independent of Y given Z if and only if the probability distribution governing X is independent of the value of Y given Z.*

$$(\forall\ i,\ j,k)P(X = x_i | Y = y_j, Z = z_k) =\ P(X = x_i | Z = z_k) \tag{2}$$

### 4.2.2. Derivation of NBA

The NBA is an algorithm of classification that uses both the Bayes rule and assumptions of conditional independence. The goal is $P(Y \mid X)$ to learn, and NBA assumes that each $X_i$ is conditionally independent of each of the other $X_k$ given Y and also independent of each subset of the other $X_k$'s given Y. This simplifies $P(X \mid Y)$.

When X contains conditionally independent $n$, the following expression is derived.

$$P(X_1 \ldots X_n) | Y) = \prod_{i=1}^{n} P(X_i | Y) \tag{3}$$

Y is any discrete valued variable, and $X_1, \ldots, X_n$ are any discrete or real valued attributes that NBA derives. The purpose is to train a classifier to produce the probability distribution across all possible Y values for each new instance X that needs classification. Following Bayes' rule, the equation for the probability that Y will take on its $k$th value is described as follows:

$$P(Y = y_k | X_1 \ldots X_n) = \frac{P(Y = y_k)\ P(X_1 \ldots X_n | Y = y_k)}{\sum_j P(Y = y_j)\ P(X_1 \ldots X_n | Y = y_j)} \tag{4}$$

where the sum is calculated over all possible $y_j$ values of Y. Considering conditionally independent $X_i$ given Y, (3) is rewritten as follows.

$$P(Y = y_k | X_1 \ldots X_n) = \frac{P(Y = y_k)\ \prod_i P(X_i | Y = y_k)}{\sum_j P(Y = y_j)\ \prod_i P(X_i | Y = y_j)} \tag{5}$$

Equation (5) is for the Naïve Bayes classifier, which is the fundamental equation.

### 4.2.3. Logistic Regression

Expressions of the learning functions in the form f: $P(Y|X)$, where $X = (X_1, X_2, \ldots, X_n)$ is any vector with discrete variables and Y is discrete-valued, is called logistic regression. If Y can take on any of the discrete values, $P(Y = y_k | X)$ is described as follows.

$$P(Y = y_k|X) = \frac{\exp(w_{k0} + \sum_{i=1}^{n} w_{ki}X_i)}{1 + \sum_{j=1}^{K-1} \exp\left(w_{j0} + \sum_{i=1}^{n} w_{ji}X_i\right)} \tag{6}$$

For $Y = y_k$, we have the following.

$$P(Y = y_k|X) = \frac{1}{1 + \sum_{j=1}^{K-1} \exp\left(w_{j0} + \sum_{i=1}^{n} w_{ji}X_i\right)} \tag{7}$$

$w_{ji}$ is the associated weight with the *j*th class $y_j$ and with $X_i$ input.

This form for $P(Y|X)$ has the advantage of leading to a simple linear classification expression.

For each continuous $X_i$, the distribution is Gaussian for each discrete value $y_k$, and it is defined by a mean and standard deviation to $X_i$ and $y_k$.

$$\mu_{ik} = E[X_i|Y = y_k] \tag{8}$$

$$\sigma_{ik}^2 = E[(X_i - \mu_{ik})^2 | Y = y_k] \tag{9}$$

The mean and variances for normal distributions can be easily calculated by the following formulas:

$$\hat{\mu}_{ik} = \frac{1}{\sum_j \delta\left(Y^j = y_k\right)} \sum_j X_i^j \delta\left(Y^j = y_k\right) \tag{10}$$

$$\hat{\sigma}_{ik}^2 = \frac{1}{\sum_j \delta\left(Y^j = y_k\right)} \sum_j \left(X_i^j - \hat{\mu}_{ik}\right)^2 \delta\left(Y^j = y_k\right) \tag{11}$$

where *j* is the *j*th training example, and d($Y = y_k$) is 1 if $Y = y_k$ and 0 otherwise.

## 5. Design and Implementation of the Developed NBA

This section presents the generation of a model and a prediction of the output variables given a set of features. ML implementation includes measurements of the EM field radiating from the two ESD generators, while current is simultaneously measured. Seven parameters of the ESD current and EM field are inputs to NBA, while the maximum values of the EM field are outputs. These data contained are in Table 2 [27,28] and are derived from the laboratory setup presented in paragraph 3.

**Table 2.** Variables used as input and output at NBA.

| Input Variables | Output Variables |
|---|---|
| charging voltage (*U*) | maximum electric field value ($E_{max}$) |
| maximum discharge current ($I_{max}$) | maximum magnetic field value ($H_{max}$) |
| current at 30 ns ($I_{30}$) | |
| current at 60 ns ($I_{60}$) | |
| rise time ($t_r$) | |
| distance (*d*) | |
| direction (*D*) | |

Specifically, hundreds of measurements were performed with each one of the two Schaffner's ESD generators (NSG-433 and NSG-438). This large number of measurements is due to the numerous parameters that can change and receive different values. The charging

voltage of the generator, the distances from the Pellegrini target, the three directions shown in Figure 3, and the current waveform parameters are listed in Tables 3 and 4.

**Table 3.** Measured EM field, versus ANN, and NBA results (NSG-433).

| | | | | | | | | | | | | | | | | | |
|---|---|---|---|---|---|---|---|---|---|---|---|---|---|---|---|---|---|
| | | | | | | **Schaffner ESD Generator NSG-433** | | | | | | | | | | | |
| | | **Varying Parameters** | | | | | | **Measured** | | **ANN** | | | | **NBA** | | | |
| ... No. | $U$ (kV) | $t_r$ (ns) | $I_{max}$ (A) | $I_{30}$ (A) | $I_{60}$ (A) | $d$ (cm) | $D$ | $E$ (kV/m) | $H$ (A/m) | $E$ (kV/m) | R.E. (%) | $H$ (A/m) | R.E. (%) | $E$ (kV/m) | R.E. (%) | $H$ (A/m) | R.E. (%) |
| 1 | +2 | 0.77 | 6.95 | 3.78 | 2.44 | 35 | A | 4.04 | 2.03 | 4.49 | 0.11 | 2.31 | 0.14 | 4.29 | 0.06 | 2.15 | 0.06 |
| 2 | +2 | 0.75 | 7.17 | 3.69 | 2.45 | 65 | A | 2.36 | 1.38 | 2.68 | 0.14 | 1.57 | 0.14 | 2.50 | 0.06 | 1.48 | 0.07 |
| 3 | +2 | 0.76 | 7.13 | 3.95 | 2.50 | 35 | C | 3.42 | 1.64 | 3.69 | 0.08 | 1.89 | 0.15 | 3.52 | 0.03 | 1.75 | 0.07 |
| 4 | +2 | 0.73 | 7.28 | 3.93 | 2.39 | 65 | C | 2.55 | 0.95 | 2.77 | 0.09 | 0.79 | 0.17 | 2.63 | 0.03 | 0.99 | 0.04 |
| 5 | +2 | 0.73 | 7.34 | 3.67 | 2.43 | 35 | D | 4.08 | 1.68 | 4.26 | 0.04 | 1.88 | 0.12 | 4.20 | 0.03 | 1.75 | 0.04 |
| 6 | +2 | 0.76 | 7.16 | 3.62 | −2.40 | 65 | D | 3.52 | 1.12 | 3.89 | 0.11 | 1.33 | 0.19 | 3.65 | 0.04 | 1.22 | 0.09 |
| 7 | −2 | 0.75 | −7.19 | −3.99 | −2.41 | 20 | A | −9.75 | −2.98 | −9.25 | 0.05 | −2.78 | 0.07 | −9.52 | 0.02 | −2.90 | 0.03 |
| 8 | −2 | 0.78 | −7.12 | −3.80 | −2.42 | 35 | A | −4.11 | −1.88 | −3.85 | 0.06 | −1.57 | 0.16 | −3.95 | 0.04 | −1.72 | 0.09 |
| 9 | −2 | 0.76 | −7.13 | −3.87 | −2.43 | 50 | A | −2.65 | −1.75 | −2.95 | 0.11 | −1.95 | 0.11 | −2.75 | 0.04 | −1.86 | 0.06 |
| 10 | −2 | 0.74 | −7.18 | −3.99 | −2.41 | 65 | A | −2.26 | −1.61 | −2.62 | 0.16 | −1.88 | 0.17 | −2.45 | 0.08 | −1.78 | 0.11 |
| 11 | −2 | 0.73 | −7.19 | −3.92 | −2.37 | 20 | C | −6.55 | −2.93 | −6.02 | 0.08 | −3.24 | 0.11 | −6.35 | 0.03 | −3.12 | 0.06 |
| 12 | −2 | 0.73 | −7.18 | −3.95 | −2.41 | 35 | C | −3.35 | −1.67 | −3.04 | 0.09 | −1.93 | 0.16 | −3.20 | 0.04 | −1.75 | 0.05 |
| 13 | −2 | 0.77 | −7.18 | −3.90 | −2.42 | 50 | C | −2.69 | −0.94 | −2.29 | 0.15 | −1.13 | 0.20 | −2.42 | 0.10 | −1.02 | 0.09 |
| 14 | −2 | 0.72 | −7.12 | −3.99 | −2.44 | 65 | C | −2.52 | −0.81 | −2.85 | 0.13 | −1.18 | 0.46 | −2.65 | 0.05 | −1.01 | 0.25 |
| 15 | +4 | 0.71 | 14.75 | 7.19 | 5.08 | 20 | A | 3.68 | 12.28 | 3.95 | 0.07 | 12.92 | 0.05 | 3.78 | 0.03 | 12.54 | 0.02 |
| 16 | +4 | 0.75 | 14.75 | 7.05 | 5.04 | 35 | A | 2.96 | 7.18 | 3.22 | 0.09 | 7.52 | 0.05 | 3.10 | 0.05 | 7.29 | 0.02 |
| 17 | +4 | 0.78 | 14.87 | 6.97 | 5.01 | 50 | A | 2.80 | 5.14 | 2.99 | 0.07 | 5.42 | 0.05 | 2.89 | 0.03 | 5.25 | 0.02 |
| 18 | +4 | 0.79 | 14.98 | 6.87 | 5.09 | 65 | A | 2.55 | 4.51 | 2.73 | 0.07 | 3.71 | 0.18 | 2.67 | 0.05 | 4.25 | 0.06 |
| 19 | +4 | 0.72 | 14.75 | 7.09 | 5.12 | 35 | C | 6.56 | 2.89 | 6.83 | 0.04 | 2.99 | 0.03 | 6.60 | 0.01 | 2.94 | 0.02 |
| 20 | +4 | 0.72 | 14.88 | 6.94 | 5.13 | 65 | C | 5.21 | 1.64 | 5.45 | 0.05 | 1.95 | 0.19 | 5.37 | 0.03 | 1.75 | 0.07 |
| 21 | +4 | 0.74 | 14.93 | 6.84 | 5.17 | 35 | D | 7.45 | 3.18 | 7.95 | 0.07 | 3.58 | 0.13 | 7.59 | 0.02 | 3.35 | 0.05 |
| 22 | +4 | 0.73 | 14.75 | 6.86 | 5.11 | 65 | D | 6.19 | 2.10 | 6.03 | 0.03 | 1.92 | 0.09 | 6.17 | 0.00 | 2.05 | 0.02 |
| 23 | −4 | 0.74 | −15.46 | −7.36 | −5.10 | 35 | A | −8.31 | −2.99 | −8.69 | 0.05 | −2.77 | 0.07 | −8.45 | 0.02 | −2.88 | 0.04 |
| 24 | −4 | 0.76 | −14.78 | −7.23 | −5.19 | 65 | A | −4.12 | −2.84 | −4.35 | 0.06 | −2.99 | 0.05 | −4.25 | 0.03 | −2.92 | 0.03 |
| 25 | −4 | 0.79 | −14.59 | −7.24 | −5.16 | 35 | C | −6.17 | −2.72 | −6.75 | 0.09 | −2.49 | 0.08 | −6.40 | 0.04 | −2.65 | 0.03 |
| 26 | −4 | 0.72 | −14.79 | −7.13 | −5.18 | 65 | C | −4.77 | −1.59 | −4.25 | 0.11 | −1.86 | 0.17 | −4.45 | 0.07 | −1.69 | 0.06 |
| 27 | −4 | 0.74 | −14.42 | −7.14 | −4.94 | 20 | D | −12.87 | −3.18 | −12.06 | 0.06 | −3.55 | 0.12 | −12.58 | 0.02 | −3.29 | 0.03 |
| 28 | −4 | 0.76 | −14.81 | −7.12 | −5.01 | 35 | D | −7.63 | −2.92 | −7.05 | 0.08 | −2.60 | 0.11 | −7.55 | 0.01 | −2.81 | 0.04 |
| 29 | −4 | 0.79 | −15.38 | −7.25 | −5.14 | 50 | D | −6.17 | −2.65 | −6.52 | 0.06 | −2.89 | 0.09 | −6.32 | 0.02 | −2.78 | 0.05 |
| 30 | −4 | 0.76 | −14.83 | −7.42 | −4.99 | 65 | D | −7.41 | −2.20 | −7.65 | 0.03 | −2.45 | 0.11 | −7.49 | 0.01 | −2.32 | 0.05 |

**Table 4.** Measured EM field, versus ANN, and NBA results for (NSG-438).

| | | | | | | | | | | | | | | | | |
|---|---|---|---|---|---|---|---|---|---|---|---|---|---|---|---|---|
| **Schaffner ESD Generator NSG-438** | | | | | | | | | | | | | | | | |
| | **Varying Parameters** | | | | | | **Measured** | | **ANN** | | | | **NBA** | | | |
| ... No. | $U$ (kV) | $t_r$ (ns) | $I_{max}$ (A) | $I_{30}$ (A) | $I_{60}$ (A) | $D$ (cm) | $d$ (cm) | $E$ (kV/m) | $H$ (A/m) | $E$ (kV/m) | R.E. (%) | $H$ (A/m) | R.E. (%) | $E$ (kV/m) | R.E. (%) | $H$ (A/m) | R.E. (%) |
| 1 | +2 | 0.72 | 7.12 | 3.89 | 2.42 | 20 | A | 5.13 | 2.65 | 5.51 | 0.07 | 2.93 | 0.09 | 5.28 | 0.03 | 2.74 | 0.03 |
| 2 | +2 | 0.72 | 6.97 | 3.67 | 2.48 | 50 | A | 2.42 | 1.34 | 2.70 | 0.12 | 1.78 | 0.33 | 2.59 | 0.07 | 1.52 | 0.13 |
| 3 | +2 | 0.73 | 7.16 | 3.98 | 2.49 | 65 | A | 5.81 | 1.09 | 6.29 | 0.08 | 1.51 | 0.39 | 5.99 | 0.03 | 1.25 | 0.15 |
| 4 | +2 | 0.74 | 7.35 | 3.82 | 2.54 | 20 | C | 5.19 | 2.91 | 5.55 | 0.07 | 2.69 | 0.08 | 5.40 | 0.04 | 2.77 | 0.05 |
| 5 | +2 | 0.75 | 7.17 | 3.95 | 2.43 | 35 | C | 2.34 | 1.87 | 2.09 | 0.11 | 1.52 | 0.19 | 2.19 | 0.06 | 1.63 | 0.13 |
| 6 | +2 | 0.79 | 7.29 | 3.75 | 2.49 | 50 | C | 2.36 | 1.39 | 2.71 | 0.15 | 1.13 | 0.19 | 2.51 | 0.06 | 1.25 | 0.10 |
| 7 | +2 | 0.72 | 7.25 | 3.89 | 2.46 | 65 | C | 5.84 | 1.09 | 6.14 | 0.05 | 1.37 | 0.26 | 5.97 | 0.02 | 1.20 | 0.10 |
| 8 | +2 | 0.74 | 7.39 | 3.79 | 2.39 | 20 | D | 5.63 | 2.66 | 5.89 | 0.05 | 2.98 | 0.12 | 5.74 | 0.02 | 2.86 | 0.08 |
| 9 | +2 | 0.77 | 7.16 | 3.93 | 2.30 | 35 | D | 2.98 | 2.21 | 3.29 | 0.10 | 2.64 | 0.19 | 3.11 | 0.04 | 2.39 | 0.08 |
| 10 | +2 | 0.79 | 7.22 | 3.67 | 2.45 | 50 | D | 3.02 | 1.73 | 3.28 | 0.09 | 1.54 | 0.11 | 3.13 | 0.04 | 1.59 | 0.08 |
| 11 | +2 | 0.77 | 6.97 | 3.95 | 2.39 | 65 | D | 2.73 | 1.34 | 2.41 | 0.12 | 1.03 | 0.23 | 2.51 | 0.08 | 1.14 | 0.15 |
| 12 | −2 | 0.72 | −7.07 | 3.82 | −2.32 | 20 | A | −5.94 | −2.95 | −6.22 | 0.05 | −3.28 | 0.11 | −6.01 | 0.01 | −3.16 | 0.07 |
| 13 | −2 | 0.71 | −7.06 | −3.99 | −2.41 | 50 | A | −2.60 | −1.31 | −2.33 | 0.10 | −1.03 | 0.21 | −2.43 | 0.07 | −1.14 | 0.13 |
| 14 | −2 | 0.75 | −7.09 | −3.84 | −2.52 | 20 | C | −2.83 | −5.87 | −2.95 | 0.04 | −6.12 | 0.04 | −2.85 | 0.01 | −5.94 | 0.01 |
| 15 | −2 | 0.78 | −7.09 | −3.65 | −2.36 | 50 | C | −2.40 | −1.21 | −2.22 | 0.07 | −1.48 | 0.22 | −2.28 | 0.05 | −1.30 | 0.07 |
| 16 | −2 | 0.74 | −7.19 | −4.12 | −2.44 | 20 | D | −4.13 | −2.91 | −4.33 | 0.05 | −3.22 | 0.11 | −4.21 | 0.02 | −3.02 | 0.04 |
| 17 | −2 | 0.72 | −7.12 | −3.93 | −2.24 | 35 | D | −3.93 | −1.82 | −3.67 | 0.07 | −1.69 | 0.07 | −3.81 | 0.03 | −1.71 | 0.06 |
| 18 | −2 | 0.72 | −7.29 | −3.98 | −2.32 | 50 | D | −3.42 | −1.42 | −3.15 | 0.08 | −1.20 | 0.15 | −3.26 | 0.05 | −1.28 | 0.10 |
| 19 | +4 | 0.72 | 14.69 | 6.98 | 5.03 | 20 | A | 11.61 | 3.39 | 12.21 | 0.05 | 3.62 | 0.07 | 11.89 | 0.02 | 3.54 | 0.04 |
| 20 | +4 | 0.73 | 14.72 | 6.89 | 5.04 | 35 | A | 7.22 | 2.88 | 7.44 | 0.03 | 2.99 | 0.04 | 7.34 | 0.02 | 2.85 | 0.01 |
| 21 | +4 | 0.78 | 14.79 | 6.72 | 5.06 | 50 | A | 4.83 | 2.34 | 4.99 | 0.03 | 2.01 | 0.14 | 4.95 | 0.02 | 2.12 | 0.09 |
| 22 | +4 | 0.74 | 14.59 | 6.58 | 4.54 | 65 | A | 3.58 | 1.92 | 3.87 | 0.08 | 1.68 | 0.13 | 3.74 | 0.04 | 1.71 | 0.11 |
| 23 | +4 | 0.72 | 14.67 | 6.84 | 4.09 | 20 | C | 9.69 | 3.47 | 9.97 | 0.03 | 3.72 | 0.07 | 9.78 | 0.01 | 3.49 | 0.01 |
| 24 | +4 | 0.78 | 14.72 | 7.18 | 4.65 | 50 | C | 4.79 | 2.45 | 5.19 | 0.08 | 2.69 | 0.10 | 4.94 | 0.03 | 2.49 | 0.02 |
| 25 | +4 | 0.75 | 14.89 | 6.39 | 4.66 | 20 | D | 10.63 | 3.24 | 11.35 | 0.07 | 3.66 | 0.13 | 11.02 | 0.04 | 3.41 | 0.05 |
| 26 | +4 | 0.73 | 14.70 | 6.81 | 4.78 | 50 | D | 5.49 | 2.69 | 5.75 | 0.05 | 2.25 | 0.16 | 5.60 | 0.02 | 2.41 | 0.10 |
| 27 | −4 | 0.76 | −14.89 | −7.32 | −4.81 | 20 | A | −12.39 | −3.36 | −11.75 | 0.05 | −3.55 | 0.06 | −12.04 | 0.03 | −3.48 | 0.04 |
| 28 | −4 | 0.77 | −14.63 | −7.31 | −4.66 | 50 | A | −5.19 | −2.26 | −4.91 | 0.05 | −2.56 | 0.13 | −5 | 0.04 | −2.46 | 0.09 |
| 29 | −4 | 0.74 | −14.79 | −7.21 | −4.05 | 20 | C | −12.27 | −3.33 | −12.57 | 0.02 | −3.56 | 0.07 | −12.37 | 0.01 | −3.41 | 0.02 |
| 30 | −4 | 0.78 | −14.92 | −7.29 | −5.01 | 50 | C | −4.47 | −2.36 | −4.09 | 0.09 | −2.02 | 0.14 | −4.35 | 0.03 | −2.22 | 0.06 |

In Figure 7, a diagram of the classification model building procedure [45], which was used in this paper, is presented. The posterior post-processing includes physical intuition into the model. For electrostatic discharges, all currents ($I_{max}$, $I_{30}$, and $I_{60}$), the rise time $t_r$, and the distances $d$ and the direction $D$ are always positive. The zero-frequency problem in Naïve Bayes is derived from (5) when there is a zero probability. If an instance in the test data set has a category that was not present during training, then it will assign it with "zero" probability and will not be able to produce predictions. This is known as zero frequency problem. It skews the entire performance of the classification. To overcome this 'zero-frequency problem' in our Bayesian environment, one was added to the count for every attribute value-class combination when an attribute value did not occur with every class value. To perform classification, Naive Bayes was extended to real-valued attributes most commonly by assuming a Gaussian distribution. This extension of Naive Bayes is the Gaussian Naive Bayes.

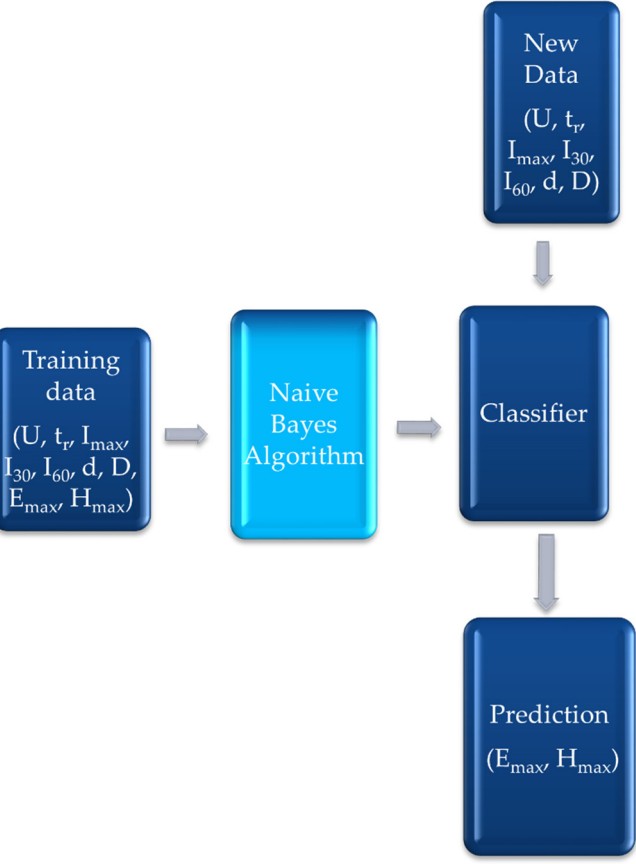

**Figure 7.** Diagram of the classification model building procedure [45].

Two sets of 1536 values of each one input and output data were utilized. These data are the measurements from the laboratory setup presented in Figure 2 and for the variables of Table 2. Twenty percent of random data were deleted from the training set in each training iteration and a validation error was calculated for these data. The training processes stopped when a root mean square error of the actual and the desired outputs (maximum values of the EM field) reached the 0.5% goal.

## 6. Results-Discussion

The results of NBA compared to the developed ANN software tool of a previous work [27,28] are presented in Tables 3 and 4. The same tables contain measurements of the EM field.

Figures 8–11 depict the percentage relative error (RE) between measured data and the ANN's or NBA's predicted values. The results obtained according to the proposed algorithm from ML are extremely close to the actual measured ones and closer than those of [27,28], proving that it has excellent function and excellent accuracy.

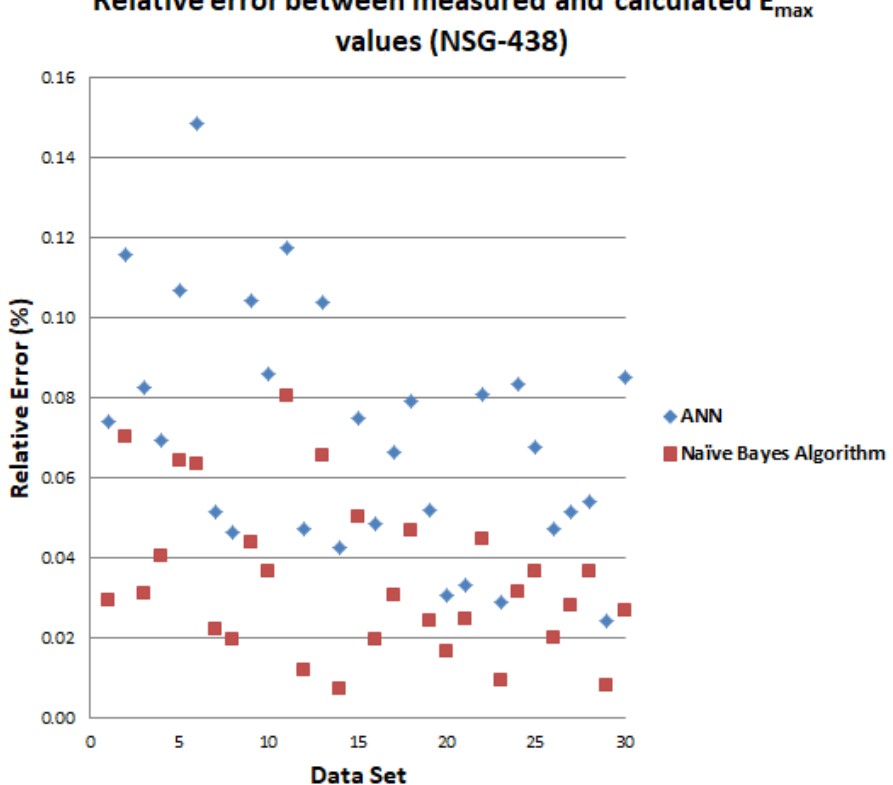

**Figure 8.** Relative error between the measured and the predicted maximum E-field of the NSG 438 using ANN and NBA.

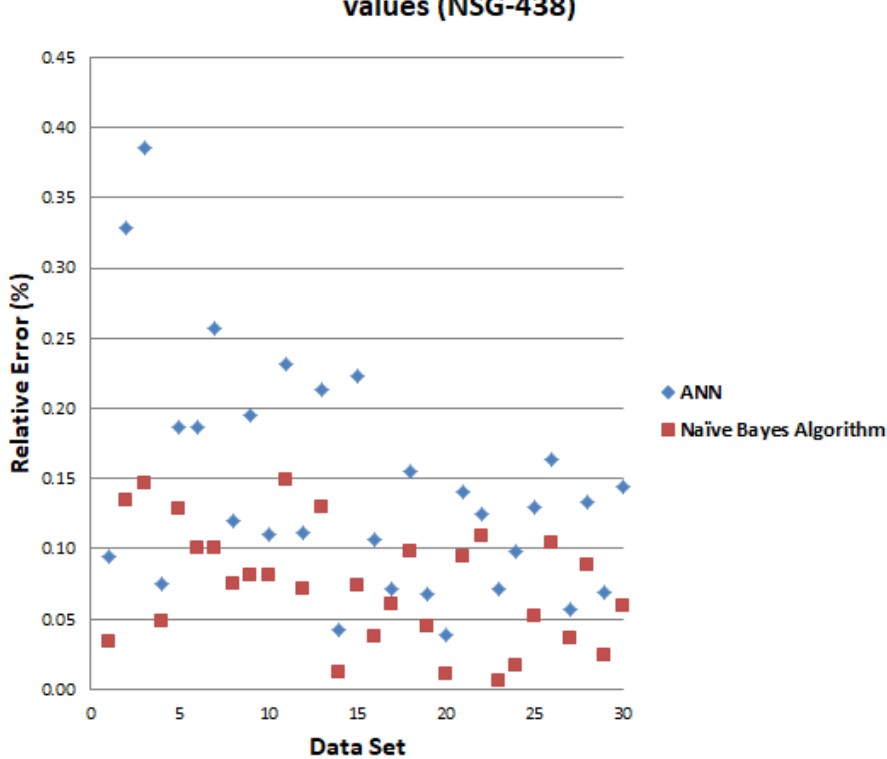

**Figure 9.** Relative error between the measured and the predicted maximum H-field of the NSG-438 using ANN and NBA.

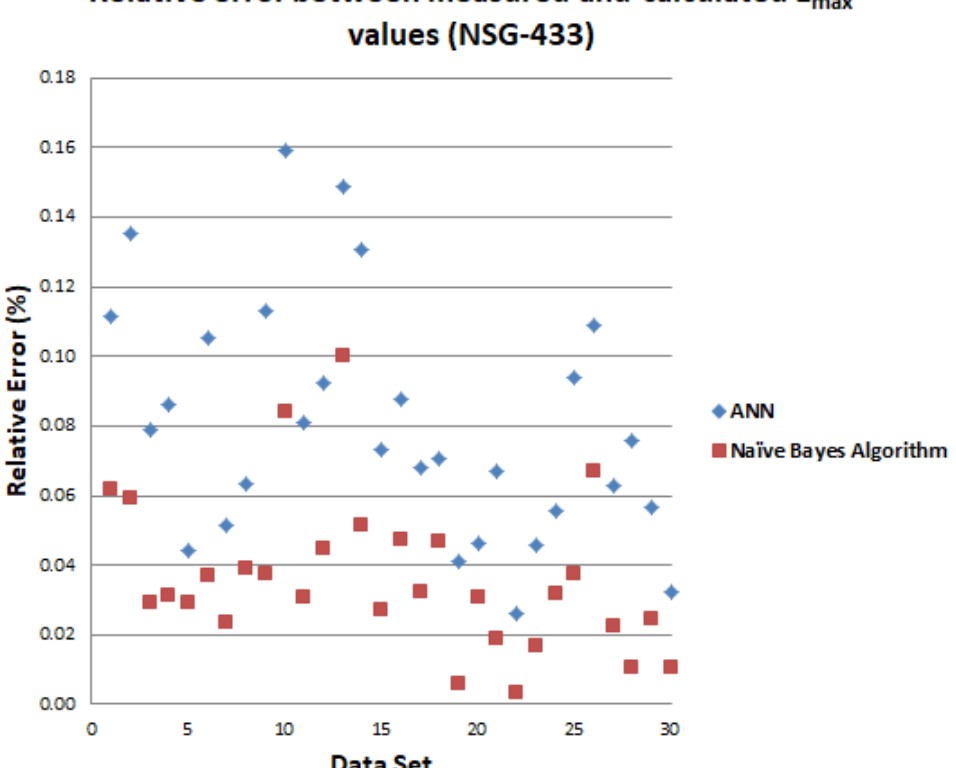

**Figure 10.** Relative error between the measured and the predicted maximum E-field of the NSG-433 using ANN and NBA.

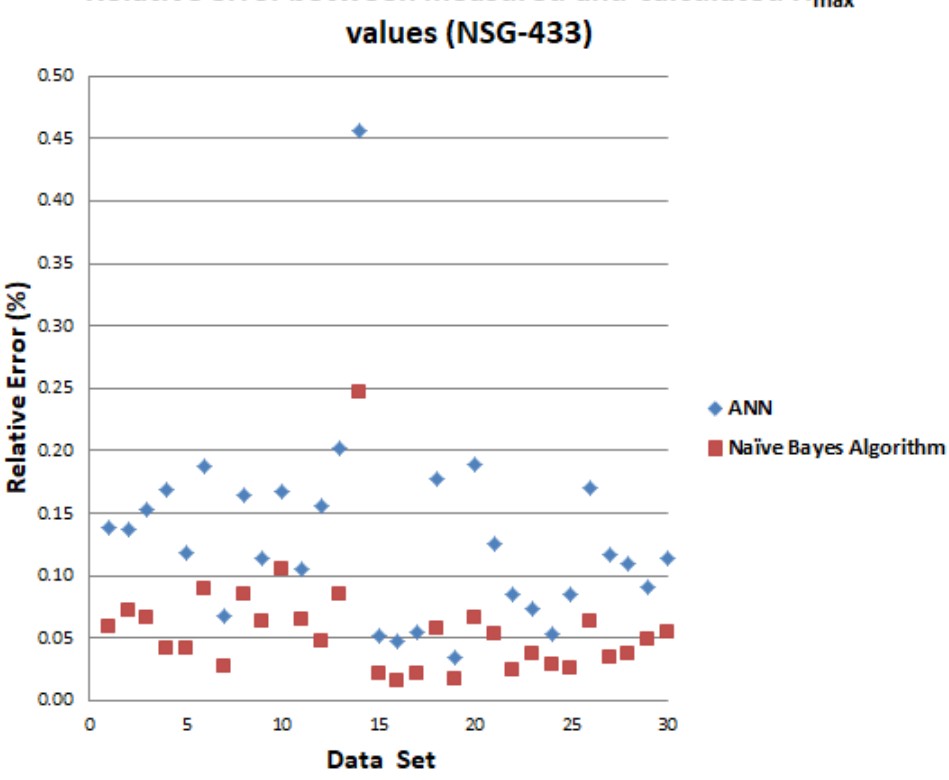

**Figure 11.** Relative error between the measured and the predicted maximum H-field of the NSG-433 using ANN and NBA.

The Naïve Bayesian algorithm is superior to ANN learning algorithms in all cases. The smallest absolute error for the NSG-433 ESD generator was 0.00% (there is a precise prediction) for the E-field that can be achieved by NBA (highlighted with yellow font color in Table 2), while for the H-field it was 0.02%. By also applying NBA for the NSG-438 ESD generator, the smallest absolute error was 0.01% both for the electric and magnetic field.

It is concluded that the predicted values of the proposed NBA both for the electric and magnetic field are extremely close to the measured ones, something that the ANN tool could not produce with such accuracy. This fact validates previous published research studies [31,32,35,36] that NBA is not only simple but also fast, accurate, and reliable. One main advantage of this algorithm is its simplicity in construction. It provides an efficient, fast, and appropriate classifier for many real-world problems.

From Figures 5 and 6, it is concluded that each ESD generator produces its own EM field. NSG-433 and NSG 438 ESD generators also radiate a different EM field depending on its orientation. Consequently, due to the different induced voltages, they produce an electronic device that is tested and may pass with one generator and fail with the other, although the discharge current is the same. Consequently, in the next revision of [11], the ESD generators should be marked with the direction in which the field is at the highest. Moreover, during ESD generator verification on the produced EM field should be tested around 360°. In such a case, the proposed NBA will predict with extreme accuracy the induced voltages on the tested devices, and it may be a useful software tool for the Electromagnetic Compatibility (EMC) laboratories that are enabled in ESD tests.

## 7. Conclusions

The paper describes the NBA from ML that assesses the EM field generated by electrostatic discharges. The applied algorithm very easily and accurately assesses produced EM field discharges by measuring the discharge current of the ESD generator and its distance and orientation from the tested equipment. The results derived by this algorithm proved its high efficiency, showing better results in the prediction of the EM field than the ANNs of a previous research study. The work of this paper has practical application for EMC Laboratories that are enabled with ESD testing and ESD generator designers as well because the proposed algorithm will be a useful tool in their work.

**Author Contributions:** Data creation, G.F.; formal analysis, G.F.; investigation, V.V.; methodology, G.F., V.V., and L.E.; supervision, L.E.; writing-original draft, G.F. All authors have read and agreed to the published version of the manuscript.

**Funding:** This research received no external funding.

**Data Availability Statement:** The data presented in this study are available upon request from corresponding author.

**Acknowledgments:** We would like to acknowledge and thank David Pommerenke of the Graz University of Technology in Austria for lending us the field sensors for the EM field measurements.

**Conflicts of Interest:** The authors declare no conflict of interest.

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
