# Peer review of "Machine Learning Techniques for the Prediction of the Magnetic and Electric Field of Electrostatic Discharges"

_electronics, doi:10.3390/electronics11121858_

Round 1
Reviewer 1 Report
This paper describes a study on the prediction of electric and magnetic field radiating by electrostatic discharges using the ML method. However, the paper is incomplete and the discussion is inadequate. From the ML point of view, it is not clear how many data sets are available in the database and what percentage of the training set and test set are used. From the discussion part of the paper, the content is too little.
1. keywords need to be revised, is ESD a commonly used keyword in the field? IEC 61000-4-2 should be removed.
2. abstract has some weaknesses; it is suggested to add one or two sentences to briefly introduce why this study is conducted.
3. introduction lacks references, especially the first two paragraphs.
4. ['learn,'] on line 147, should be revised to ['learn',].
5. section 5 and 6 should be adjusted to one section.
6. The section on data set construction is not clear and should be revised.
7. There are some data blanks in Table 3, and how the data containing these blanks are used. It is usually understood that there is a data blank and the whole row should be removed.
8. section of Results-discussion is too small; this section is the focus and should be expanded. Currently there are only about 100 words, is this the completed version.
9. Figure 8-11, regarding the y coordinate, what is the meaning of the “dot” before the numbers?
10. English needs to be improved.
Author Response
The paper has been revised according to the reviewer’s comments.
-
Keywords have been revised. ESD has been replaced by “electrostatic discharge” and IEC 61000-4-2 has been removed. Also, “contact discharges” has been added
-
Two sentences have been added at the end of the abstract that briefly explain the why this study is conducted.
-
References in the first two paragraphs of introduction have been added.
-
['learn,'] on line 147 (of the initial paper) or line 158 (of the revised paper), has been revised to ['learn',].
-
Section 5 and 6 has been adjusted to section 5. Consequently, sections 7 and 8 have been replaced by sections 6 and 7 respectively.
-
The section (lines 256-262) (of the initial paper) or 263-271 (of the revised paper), on data set construction was revised.
-
All data on Table 3 are now shown clearly. The data blanks happened accidentally due to the wrong width cell.
-
The section Results – Discussion has been expanded.
-
In Figures 8-11 the “dot” before the numbers have been removed and the number are written as decimal numbers.
-
The paper has been revised according to the reviewer’s comments. Any existing grammar mistakes have been corrected. A significant change in grammar and expressions has been made.
Reviewer 2 Report
§ Generally, for sampling in research analysis where adequate use of control experiments, accuracy of process data, and regularity of sampling have been made in time-dependent studies.
Detailed methodology is used and data analysis is conducted systematically (in qualitative research), this qualitative research extends beyond the author's opinions, with sufficient descriptive elements and appropriate quotes from interviews and focus groups.
§ The sufficient data and clear data tables.
The data consistent, also agree with the conclusions.
Confirmatory data that adds more, if anything, to current standing - unless strong arguments for such repetition are made.
§ The premise interesting and important.
§ The methods used appropriate.
§ The data support the conclusions.
§ The materials and methods used in this research follow best practices and repeatable research to obtain further results.
§ This makes effective use of the following: Control experiments - Analysis of experiments - Sampling documents more positive for this research.
§ Replicable Research
§ This makes sufficient use of:
§ Control experiments
§ Repeated analyses
§ Repeated experiments
§ Sampling.
Author Response
The authors want to express their gratitude for the reviewer comments.
Reviewer 3 Report
The paper describes the Naïve Bayes Algorithm from Machine Learning that assesses the electric and the magnetic field radiating by electrostatic discharges. Mathematical models are described. Presented a simplified diagram of the general model building procedure for pattern classification. The suggested of the Naïve Bayes Algorithm was tested.
1. To better understand the benefits of a the Naïve Bayes Algorithm from Machine Learning , in Tables 3 and Tables 4, indicate the error values between the the Naïve Bayes Algorithm and the ANN.
2. In Figures 8-11, for clarity, highlight the value of 0.00% of the relative error between measured and calculated forecasts.
Author Response
The paper has been revised according to the reviewer’s comments.
- In Tables 3 and 4 columns for the relative error of the E field and the H field, both for the NN and the Naïve Bayes Algorithm have been added.
- In Tables 3 the value of 0.00% has been highlighted.
Round 2
Reviewer 1 Report
After revision, the paper has been improved.
But there are still some minor issues that need to be carefully sculpted. For example, the background introduction directly presents machine learning without talking about what machine learning is. But in the fourth part of the paper, the author devotes a paragraph to introduce machine learning. I think this is a bit of putting the cart before the horse, and needs to be considered and revised.
Author Response
The authors following the reviewer's suggestion have included a paragrapgh at line 67 in order to introduce machine learning.